# A Newly Isolated *Alcaligenes faecalis* ANSA176 with the Capability of Alleviating Immune Injury and Inflammation through Efficiently Degrading Ochratoxin A

**DOI:** 10.3390/toxins14080569

**Published:** 2022-08-20

**Authors:** Rui Zheng, Hanrui Qing, Qiugang Ma, Xueting Huo, Shimeng Huang, Lihong Zhao, Jianyun Zhang, Cheng Ji

**Affiliations:** State Key Laboratory of Animal Nutrition, College of Animal Science and Technology, China Agricultural University, Beijing 100193, China

**Keywords:** ochratoxin A, biodegradation, *Alcaligenes faecalis* ANSA176, immune injury, inflammation, layers

## Abstract

Ochratoxin A (OTA) is one of the most prevalent mycotoxins that threatens food and feed safety. Biodegradation of OTA has gained much attention. In this study, an *Alcaligenes faecalis* strain named ANSA176, with a strong OTA-detoxifying ability, was isolated from donkey intestinal chyme and characterized. The strain ANSA176 could degrade 97.43% of 1 mg/mL OTA into OTα within 12 h, at 37 °C. The optimal levels for bacterial growth were 22–37 °C and pH 6.0–9.0. The effects of ANSA176 on laying hens with an OTA-contaminated diet were further investigated. A total of 36 laying hens were assigned to three dietary treatments: control group, OTA (250 µg/kg) group, and OTA + ANSA176 (6.2 × 10^8^ CFU/kg diet) group. The results showed that OTA decreased the average daily feed intake (ADFI) and egg weight (EW); meanwhile, it increased serum alanine aminopeptidase (AAP), leucine aminopeptidase (LAP), β2-microglobulin (β2-MG), immunoglobulin G (IgG), tumor necrosis factor-α (TNF-α), and glutathione reductase (GR). However, the ANSA176 supplementation inhibited or attenuated the OTA-induced damages. Taken together, OTA-degrading strain *A. faecalis* ANSA176 was able to alleviate the immune injury and inflammation induced by OTA.

## 1. Introduction

Ochratoxin A (OTA) is one of the most prevalent mycotoxins produced by fungi of the genus *Aspergillus* and *Penicillium* [1]. The natural contamination of OTA widely occurs in plant- and animal-derived food commodities, including cereal, fruit, coffee, meat, milk, and other agricultural food and feed [2,3]. Studies have indicated that OTA possesses carcinogenic, teratogenic, genotoxic, and immunotoxic properties [2,4,5]. The International Agency for Research on Cancer (IARC) has classified OTA as a probable carcinogen for humans (group 2B) [6]. 

In animals, OTA has a broad range of harmful effects on livestock health and productivity, resulting in food-safety incidents and economic losses [7]. Due to the high plasma-protein binding affinity and low metabolism rate of OTA, its bioaccumulation and carryover effects in edible tissues and animal-derived products further endanger human health [2,8]. For poultry’s complete feed, the European Commission Recommendation 2006/576/EC limits the maximum tolerable level for OTA to 100 µg/kg [9]. Residues of OTA are detectable in the liver, kidney, and eggs in OTA-challenged laying hens [10]. Feeding OTA contaminated diets to poultry typically causes decreased production performance, altered serum biochemical profiles, and impaired immune response [10,11]. The detrimental effects of OTA on the poultry industry have been indicated to be time- and dose-dependent, with the liver and kidney being the main target organs [4,12]. Exposure of 0.5–3 mg/kg OTA to poultry typically results in renal lesions, followed by decreases in feed consumption, grow rate, feed-conversion efficiency, egg production, and egg quality [7,12,13]. The effects of 1–5 mg/kg OTA on poultry serum biochemistry include decreased total protein (TP), albumin (ALB), and globulin levels and increased alkaline phosphatase (ALP) and gamma glutamine transpeptidase (GGT) levels [12]. Studies have been carried out for decades to elucidate the critical role of oxidative stress in OTA-induced toxicity [14]. The investigation of oxidative stress or antioxidant potential in OTA-fed (0–6.4 mg/kg) broiler chicks reveals a significant dose-dependent decrease in the superoxide dismutase (SOD), glutathione peroxidase (GPx), and total antioxidant status (TAS) in plasma and tissues [15]. 

To date, numerous physical, chemical, and biological methods have been proposed to eliminate OTA [16]. Among them, biodetoxification of OTA is a highly promising approach due to its specific, efficient, and environmentally friendly advantages [3]. Biological detoxification methods could be classified into adsorption and degradation. Many yeasts have been reported to remove OTA via high adsorption, in which cell viability is not a prerequisite and the most important influence factor is cell wall components [3,17]. *Lactobacillus rhamnosus* GG has also been found to decrease OTA during the first 15 h of culture growth [18]. However, the amount of toxin elimination could be partially reversed, indicating both a limitation and risk. The degradation of OTA to ochratoxin α (OTα: less toxic or almost nontoxic) and L-β-phenylalanine (Phe) by breaking the amide bond is considered the most important biodegradation mechanism [3,16]. A good deal of microorganisms have been identified as being able to degrade OTA, including fungi [19,20,21], bacteria [22,23], and yeast [24]. For example, *Aspergillus niger* M00120 isolated from the soil has the strong ability to detoxify 99% of OTA in 2 days [25]. Furthermore, the product has been identified as OTα and assessed for cytotoxic response, indicating that it does not induce cellular oxidative damage [25]. As for the bacteria, the strain *Acinetobacter* sp. Neg1 that was isolated from the OTA-contaminated vineyard is capable of converting 91% of OTA into OTα in six days, at 24 °C [26]. The biodegradation production has been confirmed as OTα by liquid chromatography with high-resolution mass spectrometer (LC–HRMS) [26]. However, the degradation efficiencies of some microbes might be limited, and those isolated from the gastrointestinal tract of healthy animal might be worth investigating. Notably, the application of microorganism to both the food and feed industries must be cautious of its safety. 

The objective of this study was to obtain a high-efficient OTA-degrading strain, elucidate the detoxification mechanism, and examine its protective efficacy in OTA-fed poultry. The central hypothesis is that supplementation of this novel OTA biodegrading strain could alleviate the OTA-induced immune injury and inflammation in laying hens.

## 2. Results

### 2.1. Isolation and Characterization of OTA-Degrading Strain

Colonies grown on the Luria-Bertani (LB) agar plate were screened from the donkey intestinal chyme. One pure strain, ANSA176, exhibiting the highest OTA removal ability was isolated and stored in our laboratory (Figure 1). The degradation test showed that ANSA176 was able to degrade 97.43% of OTA and produce OTα within 12 h (Figure 2), indicating the cleavage of the amide bond in OTA. The constructed phylogenetic tree (Figure 3) was consistent with the phylogeny of some *Alcaligenes faecalis* (*A. faecalis* ANSA176). Considering the microscopic observations, biochemical characteristics (Table 1), and the 16S rRNA gene sequence, the strain used in this study was confirmed and named *A. Faecalis* ANSA176. 

Furthermore, we determined the growth of ANSA176 by OD_600_ measurement. As shown in Figure 4, the optimal temperatures for growth were observed to be between 22 and 37 °C (OD_600_ = 1.73–1.87). The lower (17 °C) and upper (42 °C) temperatures limited growth to some extent. In addition, the approximate pH levels for growth were 6.0–9.0 (OD_600_ = 1.54–1.79) at 37 °C incubation. At pH 10.0, the growth was reduced. Bacterial species subjected to the tested pH ranging from 2.5 to 5.0 were unable to grow. 

### 2.2. Protection of A. faecalis ANSA176 against OTA-Induced Damages in Laying Hens

#### 2.2.1. Production Performance in Laying Hens 

In Figure 5a, we can see that the average daily feed intake (ADFI) of the OTA-fed group (2.02%, *p* < 0.05) and the OTA + ANSA176 group (6.75%, *p* < 0.05) was significantly lower than that of the control group. In the OTA + ANSA176 group, the ADFI was even lower (4.83%, *p* < 0.05) than that of the OTA-fed group. However, no statistically significant differences (*p* > 0.10) were found in the feed/egg ratio (FER, Figure 5b) when compared the OTA-fed group (an increase of 3.63%) or the OTA + ANSA176 group (a decrease of 5.16%) to the control group. Moreover, the FER was significantly decreased (8.48%, *p* < 0.05) in the OTA + ANSA176 group when compared to the OTA-fed group. During the experiment period, feeding layers with OTA at the concentration of 250 µg/kg decreased the egg production ratio (EPR; see Figure 5c; 4.61%, *p* > 0.10) and average daily egg mass (EM; see Figure 5d; 5.44%, *p* > 0.10) when compared with the control. Meanwhile, the supplementation of ANSA176 to the OTA-fed group ameliorated the negative effects by increasing EPR and EM at 0.94% (*p* > 0.10) and 2.46% (*p* > 0.10), respectively. 

The effects OTA and ANSA176 on egg weight (EW), shell percentage, yolk percentage, and albumen percentages were significantly different (*p* < 0.05) during the experimental period, while there were no statistically significant differences (*p* > 0.10) in shell color, shell thickness, shell strength, Haugh unit, and yolk color (Table 2). As shown in Table 2, the EW in the OTA group was significantly lower than the control and OTA + ANSA176 groups (*p* < 0.05). The higher shell and yolk proportions and lower albumen proportion were found in the OTA group compared to those of the control group (*p* < 0.05). Meanwhile, the supplementation of ANSA176 in the OTA-containing diet ameliorated these changes. In addition, residues of OTA an OTα were not detected (limit of detection = 0.1 µg/kg) in the eggs of all groups.

#### 2.2.2. Kidney and Liver Damage Related Parameters in Laying Hens

The administration of OTA resulted in mild pathomorphological changes in kidney and liver. Histopathological examination revealed that OTA caused kidney tubulonephrosis with degeneration of the epithelial cells in proximal tubules and glomerulonephrosis with enlarged glomeruli and swollen capillary endothelial cells (Figure 6a). In addition, OTA induced inflammatory infiltration and hyaline degeneration in the liver, which were ameliorated in the OTA + ANSA176 group (Figure 6b).

As shown in Figure 6c,d, the serum alanine aminopeptidase (AAP) (*p* < 0.05) and leucine aminopeptidase (LAP) (*p* = 0.05) levels increased in the OTA group compared to the control group, whereas the supplementation of ANSA176 obviously decreased (*p* < 0.05) them. The concentration of serum alanine aminotransferase (ALT) was significantly higher (*p* < 0.05) in the OTA + ANSA176 group than in the control and OTA groups. No differences (*p* > 0.10) were observed in phosphoenolpyruvate carboxykinase (PEPCK), creatinine (Cr), aspartate aminotransferase (AST), and alkaline phosphatase (ALP) among the three groups (Figure 6e,f,h,i). 

#### 2.2.3. Immune and Inflammatory Response in Laying Hens 

The immune and inflammatory response of OTA and ANSA176-fed layers are presented in Figure 7. In the serum, the levels of β2-microglobulin (β2-MG) (*p* < 0.05), immunoglobulin G (IgG) (*p* = 0.06), and tumor necrosis factor-α (TNF-α) (*p* = 0.05) after OTA feeding showed increases compared with the control group. Meanwhile, the concentrations of IgG (*p* < 0.05), lysozyme (LZM) (*p* < 0.05), interleukin-2 (IL-2) (*p* < 0.05), and TNF-α (*p* = 0.09) in the OTA + ANSA176 groups were lower than the OTA-fed group. There were no obvious differences (*p* > 0.10) in total protein (TP), albumin (ALB), IgA, IgM, and IL-10 among all groups.

#### 2.2.4. Oxidative Stress and Antioxidant Status in Laying Hens

Figure 8 exhibits the oxidative stress and antioxidant status in laying hens caused by the OTA and ANSA176 diet. The serum total antioxidant capacity (T-AOC) and superoxide dismutase (SOD) levels were significantly higher (*p* < 0.05) in the OTA + ANSA176 group compared with other groups. The glutathione reductase (GR) level was significantly higher (*p* < 0.05) in the OTA-fed group than in the control group. The levels of malonaldehyde (MDA), total glutathione (T-GSH), and glutathione peroxidase (GSH-Px) differed non-significantly (*p* > 0.10) among all groups.

## 3. Discussion

Ochratoxin A is a toxic secondary fungal metabolite that widely contaminates agriculture products, thereby threatening animal and human health. Microorganisms with an OTA-biodegradation property have great administration prospects in the food and feed industries due to their advantages in high specificity and effectivity [3]. In this study, we isolated and identified the *A. faecalis* ANSA176 strain from donkey intestinal chyme and found that it is capable of degrading 97.43% of OTA to OTα within 12 h in vitro. In addition, ANSA176 showed optimal growth at 22–37 °C and pH 6.0–9.0. Taken together, these results indicated a potential application of ANSA176 for the OTA biodegradation use due to its easy culture and high degradation efficiency. Microorganisms with a confirmed ability to transform OTA into OTα have been previously studied [3,27]. It is noteworthy that an *A. faecalis* ASAGF 0D-1 strain isolated from soil and a novel N-acyl-L-amino acid amidohydrolase cloned from *A. faecalis* DSM 16503 have been reported to degrade OTA into OTα [23,28]. The biodegradation of OTA into barely or non-toxic metabolites (OTα and Phe) by microorganisms and their intracellular or extracellular enzymes is a preferred method to eliminate the toxic and carcinogenic potential. In many cases, biodegradation entailed the use of enzymes, including crude and purified ones that were able to cleave OTA [3,16]. In recent years, carboxypeptidases from *Bacillus amyloliquefaciens* ASAG1 [29], *Acinetobacter* sp. neg1 ITEM 17016 [30], and *Bacillus subtilis* ANSB168 [31] have been cloned and expressed to hydrolyze OTA. Except for the well-studied carboxypeptidase, several other enzymes were also indicated to biodegrade OTA, including protease A [32], amidohydrolase [33], lipase A [34], and hydrolase [35]. However, the degradation efficiencies of some microbes might be limited. Isolating strains from the gastrointestinal tract of healthy animals might be worth investigating. Our assay may help shed some light on investigating highly efficient enzymes. Further purification and identification of the degradation enzymes involved in OTA degradation by the strain ANSA176 is still ongoing. 

The exposure of OTA could cause safety issues and enormous economic losses. Meanwhile, the efficacy and melioration effects of microorganism on OTA challenged poultry are still limited [3]. A study of varying levels of OTA showed that 2 mg/kg OTA decreased EPR and increased FER, while 4 mg/kg OTA decreased shell thickness [36]. Denli et al. found that the 2 mg/kg OTA-contaminated diet significantly decreased EM compared to the control [11]. Moreover, OTA at a level of 0.25 mg/kg did not influence the number of daily eggs produced, but 1 mg/kg of OTA reduced that, indicating the dose-dependent effect [13]. In order to provide a theoretical basis for the practical application of *A. faecalis* ANSA176, we further analyzed its alleviation effects on the laying performance of OTA-fed hens. In this experiment, decreases in ADFI and EW were recorded in the OTA group, which were in line with other studies [11,36]. Notably, administration of ANSA176 obviously ameliorated OTA-induced effects on layer’s EM and FER. In addition, the OTA changed shell, yolk, and albumen percentages were altered back toward the control’s by ANSA176. The unexpected ADFI decline in the ANSA176 supplementation group could be due to the palatability change. Although little is known on the palatability effects of bacteria supplements in feed diet, it has been accepted that microbial fermentation could change palatability and consumption [37,38]. In addition, a pair-feeding model indicated that the adverse effects of mycotoxin on growth performance, antioxidative status, and inflammation reaction were more severe compared to the merely FI reduction [39].

The challenge of OTA has been shown to result in nephrotoxic, hepatotoxic, and immunotoxic issues [14]. Several biochemical indicators in the serum can be used to reflect the tissue damage and inflammatory status in animals. Tubular enzymes AAP and LAP were markers of nephron injury [40]. The high β2-MG level was considered a clinical marker for nephropathy, and its excretion was related to OTA in Balkan endemic nephropathy (BEN) [41]. Our results showed increases in serum AAP, LAP, and β2-MG concentrations in OTA-fed layers, indicating OTA-induced damage. In comparison, biodegradable ANSA176 significantly reduced AAP and LAP, revealing a mitigating effect. Antibody responses are initiated via immune cells and lead to the production of immunoglobulins. It has been reported that OTA-contaminated diet significantly increased IgA, IgG, and IgM, suggesting an immune response against mycotoxin [31]. Cytokines such as IL-2 and TNF-α were involved in systemic inflammation and immunity [42,43]. Al-Anati et al. demonstrated that OTA released TNF-α in a dose- and time-dependent fashion via the classical inflammatory signaling cascade [44]. In the current study, increases were observed in serum β2-MG, IgG, and TNF-α concentrations in layers exposed to dietary OTA, indicating the OTA induced immune response and inflammation. However, the supplementation of ANSA176 can reduce levels of IgG, LZM, IL-2, and TNF-α to present protective effects. 

Oxidative stress and antioxidant potential are sensitive indicators reflecting the imbalance between systemic reactive oxygen species (ROS) and the biological ability to detoxify intermediates or repair damage [14,15]. Nowadays, the various detrimental effects of OTA have been associated with the toxin itself, as well as the influence of ROS and oxidative stress [14]. Both in vitro and in vivo studies have suggested that the OTA-induced oxidative stress and reduced antioxidant ability may be implicated in the renal toxicity and carcinogenicity [14,39,45]. Different antioxidant enzymes and total antioxidant status (TAS) reflected the disturbances in animals [15]. Considering that OTA enhances the production of free radicals, the activity of antioxidant enzymes such as SOD may be affected. A previous study showed that feeding OTA to broiler chicks resulted in a dose-dependent reduction of TAS and SOD in plasma and almost all the tissues [15]. Although limited information was available on the OTA-induced oxidative stress in poultry, studies on rats have been carried out [14]. The enzymatic antioxidant SOD was significantly decreased in the kidney of OTA-challenged rat, while the administration of alleviators restored it [46,47]. In addition, pretreatment of antioxidant inhibited OTA-induced ROS overproduction and SOD reduction [48]. Furthermore, the injection of antioxidants SOD and catalase provided protection against OTA in rats [49]. In the present study, levels of T-AOC and SOD were higher in the ANSA176-treated group, suggesting an ameliorative effect of oxidative stress inside layers. It also supported the hypothesis that the modulation of oxidative stress might be due to the change in antioxidant enzyme, which has been proposed before [15,50]. 

## 4. Conclusions

In summary, our findings showed that the OTA-biodegrading strain *A. faecalis* ANSA176 could be used as a potential bioproduct to alleviate OTA-induced kidney and liver damage, inflammation, and oxidative stress. The application of the ANSA176 in grain or feed needs to be further developed, especially in resolving the consumption decline. Studies on the identified OTA degradation enzymes from ANSA176 also have a promising future.

## 5. Materials and Methods

### 5.1. Isolation and Identification of the OTA-Degrading Strain 

#### 5.1.1. Culture Media and Reagents

The LB broth containing 10 g/L peptone, 5 g/L yeast extract, and 10 g/L NaCl was used for bacteria growth. All the reagents used in this study were purchased from Sigma-Aldrich (St Louis, MO, USA), unless otherwise indicated. The OTα standard was purchased from Biopure, Romer Labs (Tulln, Austria).

#### 5.1.2. Isolation of Microorganisms with OTA Degradation Ability

Bacterial strains were isolated from the intestine of animals. About 0.05 g of intestinal content was added to 5 mL of phosphate saline buffer and mixed by vortex. The solutions were further diluted and cross-streaked in Petri dishes containing LB agar to generate individual colonies of bacteria. Then the colonies obtained were individually transferred to LB broth and incubated on a rotary shaker (*n* = 200 rpm) at 37 °C for 24 h. The cultures were obtained through two successive inoculations (inoculum size at 2%, *v*/*v*). An aliquot of 980 µL of the culture was mixed with 20 µL of 50 mg/mL OTA standard solution. Sterile LB medium was used as a control. The inoculated mixtures were incubated at 37 °C at 200 rpm for 24 h. An equal volume of methanol (1 mL) was added at the end of incubation.

#### 5.1.3. Determination of OTA and OTα by Using HPLC

After incubation, cells of microbes were removed by centrifugation at 12,000 rpm for 4 min. The supernatants were filtered through filters with a 0.22 µm pore size (Millipore, Temecula, CA, USA) and analyzed by using high-performance liquid chromatography (HPLC). The HPLC was performed on an instrument (RF-20A, SHIMADZU Corporation, Kyoto, Japan) that contained a fluorescence detector set at 333 nm excitation and 477 nm emission wavelengths. The analysis was carried out with a reversed-phase Agilent Eclipse Plus C18 column (4.6 × 150 mm, 5 µm) at an injection volume of 20 µL, and the mobile phase was water/acetonitrile/acetic acid (99:99:2, *v*/*v*/*v*) at a flow rate of 1.0 mL/min. During the HPLC analysis of OTA with fluorescence detection, the peak of OTα production was also clearly observed.

#### 5.1.4. Identification of the OTA-Degrading Strain

The morphological observation of the strain named ANSA176 with high OTA-degrading ability was carried out. To identify the bacterial strain, the 16S rRNA gene sequence was amplified by using primer set 27f-1492r. The inspection and appraisal report were given by the Institute of Microbiology (Chinese Academy of Sciences, Beijing, China). The strain was preserved in 20% glycerol and stored at −20 °C. Bacteria were revived and grown in LB broth at 37 °C for 24 h, with shaking (*n* = 200 rpm), for experiment. 

#### 5.1.5. Characterization of *A. faecalis* ANSA176 Growth with Different Temperature and pH Levels

In order to determine the growth of ANSA176 under different conditions, a range of temperature (17, 22, 27, 32, 37, and 42 °C) and pH (2.5, 3, 4, 5, 6, 7, 8, 9, and 10) levels were investigated during a period of 24 h. Optical densities (ODs) were measured and recorded at 600 nm. Subsequently, the number of bacteria was calculated from the equation of plate counts against OD and expressed as log_10_ CFU/mL.

### 5.2. Animal Trial in Layer Hens

#### 5.2.1. Dietary Treatments of Animal Trial

According to Qing et al. [31], an *Aspergillus ochraceus* (CGMCC 3.4412) strain was used to produce OTA by artificial infection of sterile maize for 21 days, at 25–28 °C, and then the maize was dried and smashed. The concentration of OTA in maize powder was measured by HPLC, which was later added into the basal diet at 18% to meet the predicted concentration and verified by HPLC (predicted = 250 μg/kg; measured = 247.8 μg/kg). This dose was intended to observe the adverse effect of OTA contamination and the alleviation effect of *A. faecalis* ANSA176 supplementation.

The broth of strain *A. faecalis* ANSA176 was transformed into a freeze-dried powder. The number of bacteria in the powder was later determined as 3.1 × 10^8^ CFU/g. Then it was added to the contaminated basal diet up to an overdose of 2 kg/T feed to ensure the effectiveness of degradation.

#### 5.2.2. Design of Animal Experiments

All procedures were reviewed and approved by the Laboratory Animal Welfare and Animal Experimental Ethical Committee of China Agricultural University (No. AW 13301202-1-7). The trial strictly complied with the standard operating procedures for experimental animals of the Ministry of Science and Technology (Beijing, China), and every effort was made to minimize suffering.

A total of 36 Jingfen No.1 layers (26 weeks of age) were randomly allocated to three feeding treatments and 12 replicate pens per treatment (separately feeding): (1) basal diet group (A) fed the basal diet without OTA contamination; (2) OTA-contaminated group (B) fed the contaminated diet containing exceeded limit dose of OTA (about 250 µg/kg); and (3) biodegradable agent group (C) fed the OTA contaminated diet added 2 kg *A. faecalis* ANSA176 freeze-dried culture per ton (6.2 × 10^8^ CFU/kg diet of ANSA176). For the nutritive values and feeding procedures, we referred to the NY/T 33-2004 (China) and recommendations for Jingfen No.1 layers (Huadu yukou, Beijing, China). The composition and nutrient levels of the basal and contaminated diets are shown in Appendix A
Table A1. The feeding trial lasted 70 days. On the final day of the feeding trial, blood samples were collected from each layer via the wing vein. After that, layers were euthanized for tissue collection, following the sodium pentobarbital injection (0.4 mL/kg BW; Sile Biological Technology Co., Ltd., Guangzhou, China). 

#### 5.2.3. Laying Performance and Egg Quality

The laying performance was recorded and calculated. At the end of the experiment, 30 eggs were randomly selected from each treatment group to assess the egg quality, as previously described [51]. Shell color was determined with a QCR color reflectometer (QCR SPA, TSS, York, England). Thickness and strength were tested by the eggshell tester (ESTG-01, Orka Technology Ltd., Ramat Hasharon, Israel). Haugh unit and yolk color were measured by a multifunctional egg quality tester (EA-01, Orka Technology Ltd., Ramat Hasharon, Israel). Then the yolk was separated with a separator and weighed. The relative shell, yolk, and albumen proportions were calculated. 

#### 5.2.4. Residues of OTA in Eggs

Residues of OTA and OTα in eggs were determined weekly during the animal trial, as previously conducted [31]. Briefly, homogenized sample was weighed and added into an acetonitrile/water solution (60:40, *v*/*v*). The mixture was shaken, followed by filtering to get the supernatant. The cleanup step was performed by passing the extracted sample through the immunoaffinity column (OchraTestWB, VICAM, Watertown, MA, USA) at a rate of 1 or 2 drops per second, under gentle vacuum pressure. Then the column was washed with 10 mL of water–methanol (90:10, *v*/*v*) and dried under nitrogen gas (N_2_) for 5 min. Finally, OTA was eluted by 2 mL of methanol for HPLC analysis. The HPLC parameter settings were as described above. The determined limit of detection (LOD) for OTA and OTα was0.1 µg/kg (based on a signal–noise ratio of 3). 

#### 5.2.5. Blood Sampling and Serum Biochemical Analysis

Blood samples were centrifuged at 3000 rpm for 15 min at room temperature, and then serum samples were stored at −20 °C until analyzed. Serum contents of AST, ALT, ALP, PEPCK, Cr, TP, ALB, LZM, T-AOC, SOD, and GR were measured by using diagnostic kits (Nanjing Jiancheng Bioengineering Institute, Nanjing, China), according to the manufacturer’s instructions. Serum activities of LAP, AAP, MDA, T-GSH, GSH-Px, globulins (β2-MG, IgA, IgG, and IgM), and cytokines (TNF-α, IL-2, and IL-10) were determined by ELISA kits (Nanjing Jiancheng Bioengineering Institute, Nanjing, China), following the manufacturer’s instructions.

#### 5.2.6. Liver and Kidney Histological Assessment

Part of the liver and kidney samples were fixed in 10% neutral-buffered formalin solution and embedded in paraffin. Then these sections were stained with hematoxylin and eosin for histopathological examination, as previously described [52].

### 5.3. Statistical Analysis

The experimental data were analyzed through one-way analysis of variance (ANOVA), using the Statistical Analysis System software (SAS) package version 9.4. Tukey multiple comparison analysis was performed to determine significance of differences among treatment means. Differences were considered significant if the *p*-value was < 0.05 and demonstrated a trend if the *p*-value was < 0.10. The GraphPad Prism 9 software was used to generate graphs.

## Figures and Tables

**Figure 1 toxins-14-00569-f001:**
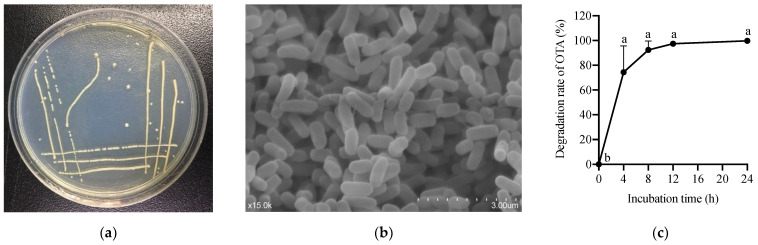
Isolation of the OTA-degrading strain. (**a**) Colony morphology. (**b**) Cell morphology. (**c**) Degradation of OTA by ANSA176 during a 24 h period. ^a–b^ The different letters in 1c mean significant difference (*p* < 0.05).

**Figure 2 toxins-14-00569-f002:**
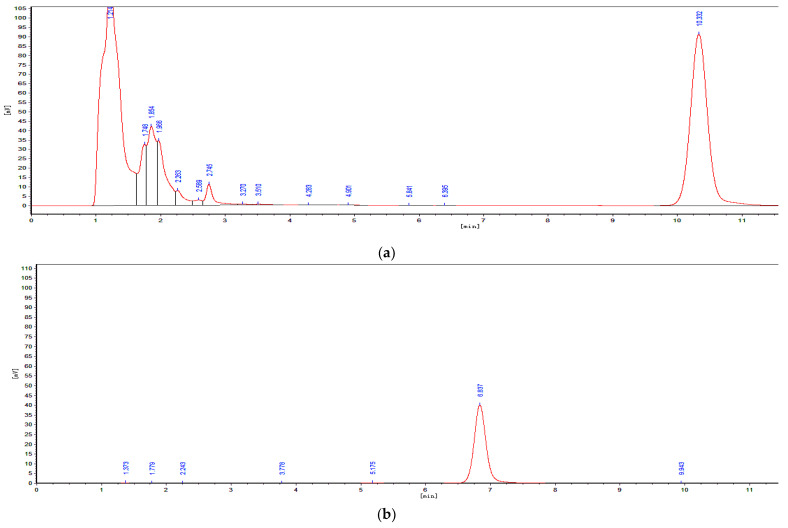
Determination of OTA degradation by HPLC. (**a**) LB culture medium with OTA standard. (**b**) OTα standard. (**c**) 30 % of OTA was degraded by ANSA176. (**d**) 75 % of OTA was degraded by ANSA176. (**e**) OTA was completely degraded by ANSA176.

**Figure 3 toxins-14-00569-f003:**
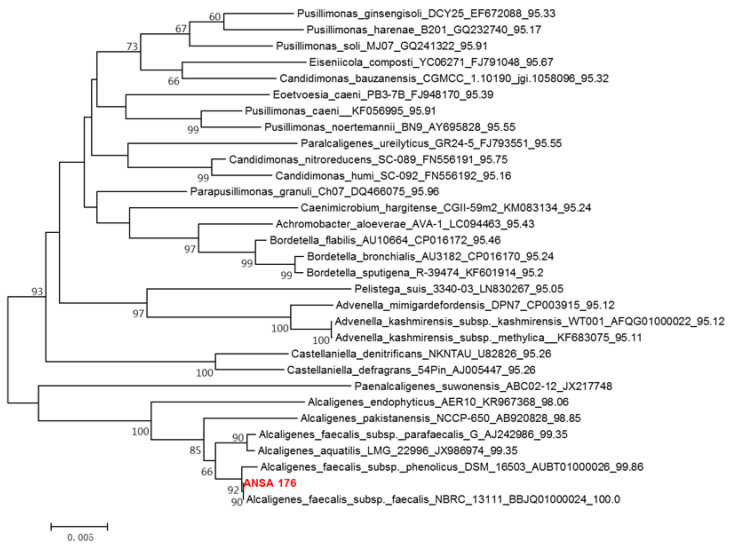
Phylogenetic tree of the isolated ANSA176 and related taxa.

**Figure 4 toxins-14-00569-f004:**
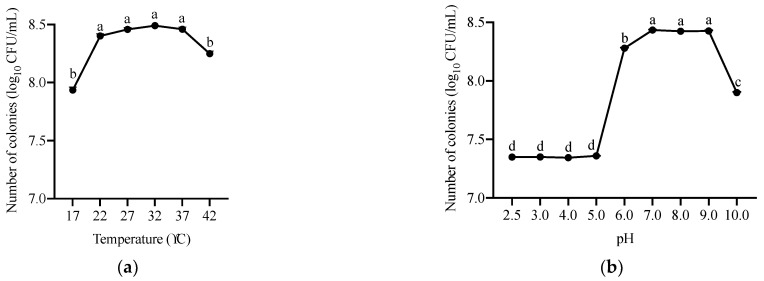
Growth characteristics of ANSA176. (**a**) Growth at different temperatures during a 24 h period. (**b**) Growth at different pH during a 24 h period. Data are presented as means ± SEM. ^a–d^ The different letters mean significant difference (*p* < 0.05).

**Figure 5 toxins-14-00569-f005:**
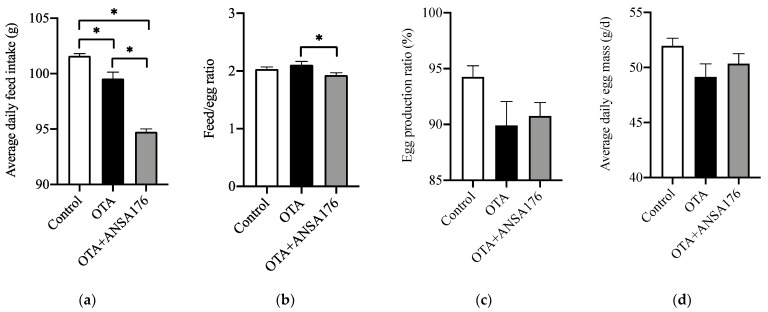
Effects of OTA and ANSA176 on production performance of layers. (**a**) Average daily feed intake. (**b**) Feed/egg ratio. (**c**) Egg production ratio. (**d**) Average daily egg mass. Data are presented as means ± SEM. The differences are defined as * *p* < 0.05.

**Figure 6 toxins-14-00569-f006:**
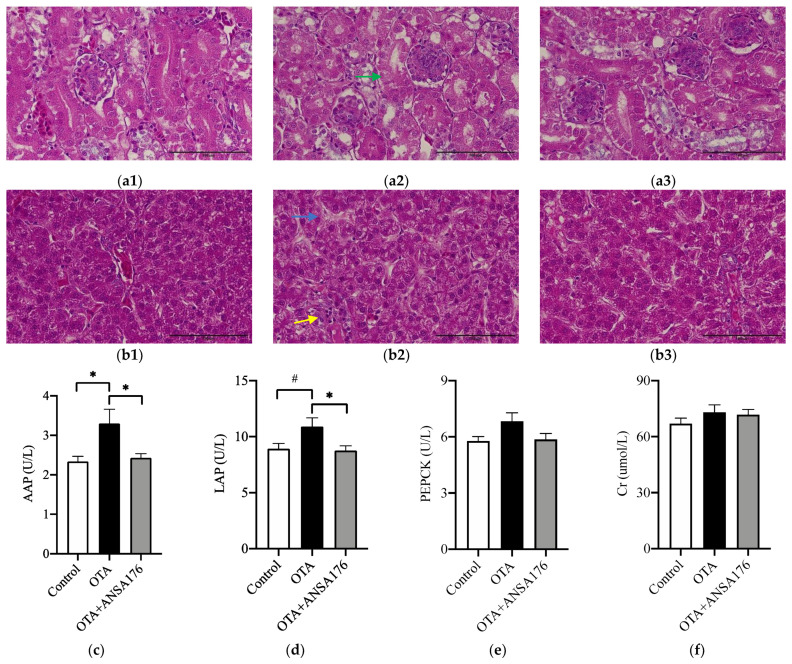
Effects of OTA and ANSA176 on kidney and liver damage related parameters in layers. (**a**,**b**) Histopathological examination of kidney (**a1**, control; **a2**, OTA; **a3**, OTA + ANSA176) and liver (**b1**, control; **b2**, OTA; **b3**, OTA + ANSA176) tissues by H&E staining. Magnification times, 400 × (scale bars = 100 µm). Lesions such as epithelial cells degeneration (green arrow), inflammatory infiltration (yellow arrow), and hyaline degeneration (blue arrow) are shown with arrows. (**c**–**i**) Serum biochemical parameters. Data are presented as means ± SEM. The differences are defined as # 0.05 ≤ *p* < 0.10 and * *p* < 0.05.

**Figure 7 toxins-14-00569-f007:**
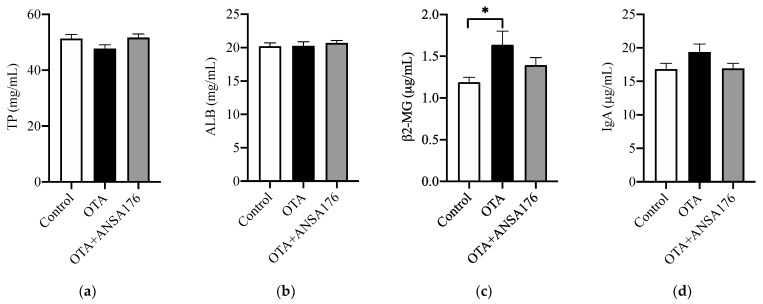
Effects of OTA and ANSA176 on immune and inflammatory response of layers. (**a**–**j**) Levels of TP, ALB, β2-MG, IgA, IgG, IgM, LZM, IL-2, IL-10, and TNF-α. Data are presented as means ± SEM. The differences are defined as # 0.05 ≤ *p* < 0.10 and * *p* < 0.05.

**Figure 8 toxins-14-00569-f008:**
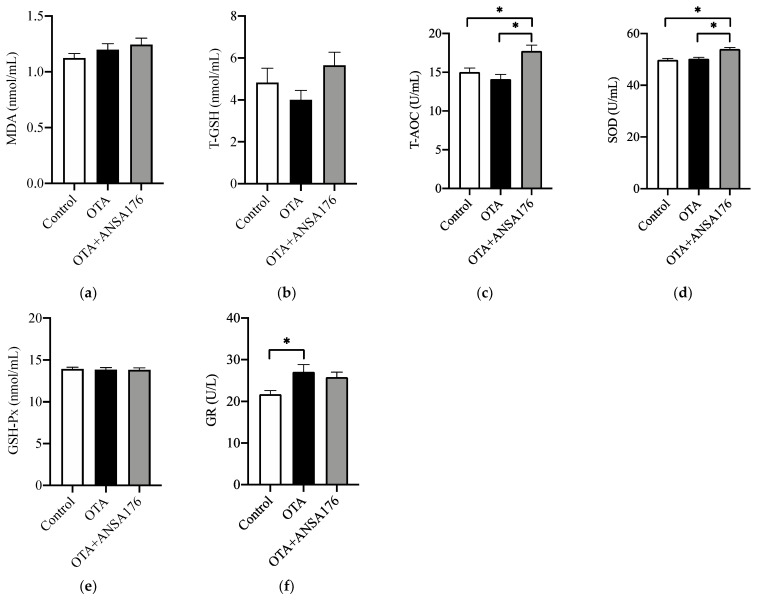
Effects of OTA and ANSA176 on oxidative stress and antioxidant status of layers. (**a**–**f**) Levels of MDA, T-GSH, T-AOC, SOD, GSH-Px, and GR. Data are presented as means ± SEM. The differences are defined as * *p* < 0.05.

**Table 1 toxins-14-00569-t001:** Biochemical and physiological characteristics of *A. faecalis* ANSA176.

Substrate/Test	Result	Substrate/Test	Result	Substrate/Test	Result	Substrate/Test	Result
Cell shape	Rod	Gram Staining	−	Catalase	+	Oxidase	+
BIOLOG GEN III (carbon source utilization assays)							
Negative control	−	α-D-glucose	−	Gelatin	−	D-raffinose	−
3-methyl-D-glucose	−	D-mannose	−	Gentiobiose	−	α-D-lactose	−
D-glucose-6-PO_4_	−	D-fructose	−	L-alanine	+	D-melibiose	−
Glucuronamide	+	D-galactose	−	L-arginine	−	L-lactic acid	+
D-cellobiose	−	Dextrin	−	L-glutamic acid	+	Citric acid	+
Glycyl-L-proline	−	L-rhamnose	−	L-histidine	+	α-ketoglutaric acid	+
L-aspartic acid	+	Inosine	−	L-pyroglutamic acid	+	D-malic acid	−
p-hydroxyphenyl acetic acid	+	D-sorbitol	−	L-serine	−	L-malic acid	+
Methyl pyruvate	+	D-mannitol	−	Pectin	−	Bromo succinic acid	+
D-lactic acid methyl ester	−	D-arabitol	−	Sucrose	−	γ-aminobutyric acid	−
β-methyl-D-glucoside	−	Myo-inositol	−	Turanose	−	α-hydroxybutyric acid	+
β-hydroxy-D,L-butyric acid	+	Glycerol	−	D-gluconic acid	−	D-salicin	−
N-acetyl-D-glucosamine	−	D-maltose	−	Stachyose	−	α-keto-butyric acid	+
N-acetyl-β-D-mannosamine	−	D-trehalose	−	Mucic acid	−	Propionic	+
N-acetyl-D-galactosamine	−	D-aspartic acid	−	Quinic acid	−	Acetic acid	+
N-neuraminic acid	−	D-serine	+	Saccharic acid	−	Formic acid	+
BIOLOG GEN III (chemical sensitivity assays; +, not sensitive; −, sensitive)							
Positive control	+	Tetrazolium blue	+	pH 6.0	+	Nalidixic acid	−
Na bromate	−	Fusidic acid	+	pH 5.0	+	LiCl	+
Rifamycin SV	+	D-serine	+	1% Na-lactase	+	K-tellurite	−
Minocycline	+	Lincomycin	+	1% NaCl	+	Aztreonam	+
Vancomycin	+	Guanidine HCl	+	4% NaCl	+	Na-butyrate	+
Niaproof 4	+	Tetrazolium violet	+	8% NaCl	+		

“+” and “−” represent the positive and negative response, respectively.

**Table 2 toxins-14-00569-t002:** Effect of OTA and ANSA176 on the egg quality of layer hens.

Treatment	Control	OTA	OTA + ANSA176	SEM ^1^	*p*-Value
Egg weight (g)	54.95 ^a^	46.64 ^b^	55.08 ^a^	1.20	<0.01
Shell percentage (%)	9.92 ^b^	12.17 ^a^	10.23 ^b^	0.30	<0.01
Yolk percentage (%)	25.90 ^b^	31.81 ^a^	25.35 ^b^	1.07	<0.01
Albumen percentage (%)	64.18 ^a^	56.01 ^b^	64.43 ^a^	1.31	<0.01
Shell color	58.03	58.01	57.69	1.31	0.98
Shell thickness (mm)	0.41	0.41	0.41	0.00	0.37
Shell strength (N)	40.75	38.77	39.90	1.12	0.46
Haugh unit	83.87	81.14	82.93	1.37	0.36
Yolk color	5.37	5.33	4.93	0.16	0.12

^1^ Pooled standard error of the mean. ^a,b^ Means with different letters within a row mean significant difference (*p* < 0.05).

## Data Availability

The data presented in this study are available on request from the corresponding author.

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
