# Peer review of "A Newly Isolated Alcaligenes faecalis ANSA176 with the Capability of Alleviating Immune Injury and Inflammation through Efficiently Degrading Ochratoxin A"

_toxins, 2022, doi:10.3390/toxins14080569_

Round 1
Reviewer 1 Report
This manuscript reports a study dealing with the potential of Alcaligenes faecalis strain to alleviate the adverse effects of OTA on the immune and inflammation in layers. There are however some critical points to be addressed:
Introduction
The introduction did not clearly present the hypothesis of this study.
Results
Figure 4 and 7. “The differences were defined as # 0.05 ≤ P< 0.10 and * P < 0.05.” There is no trend in this figure. I suggest remove this information.
Discussion
The Discussion section can be improved. I recommend that the discussion should be rewritten to be closer to the variables the authors tested in order to explain better the results and show some molecular mechanism.
L.169 Other phrase that was not in accordance is in which the authors use non-statistical differences observed in FER to explain the results.
L 187-188. SOD isn’t only one enzyme defense against ROS. I suggest rewritten this sentence.
Material and methods
The dose of OTA needs to be discussed. Is this a practical relevant dose? You have written that the maximum tolerable level in poultry feed is 100 μg/kg OTA. Why did you decide to use twice exceeding the recommended norms in the study? L240
L. 256 State method of euthanasia and blood taken (the method of blood taken may affect serum biochemical indicators, especially antioxidant enzymes).
Line 296: In the phrase: “All nutrient levels were calculated.”, did you formulate using the table data or any equation proposed by NRC ? The concentration of ME is very low.
References
According to the Journal Style is necessary to write abbreviated Journal Name in references, for example L310, 314, 327
The references nr 27 and 28 are not cited in the manuscript.
Reviewer 2 Report
The authors present the results of a study regarding ochratoxin A (OTA) biodegradation by an Alcaligenes faecalis strain named ANSA176. The authors indicated that this donkey isolated bacteria strain has strong OTA-detoxifying ability. In fact, they present data where the bacteria could degrade 97.43 % of 1 mg/mL OTA into OTα within 12 h at 37 °C with optimal growth levels at 22 – 37 °C and pH 6.0 – 9.0. In vivo experiments on the effects of ANSA176 on laying hens with OTA-contaminated diet revealed that OTA decrease in average daily feed intake (ADFI) and egg weight (EW), and increase at the serum alanine aminopeptidase 14 (AAP), leucine aminopeptidase (LAP), β2-microglobulin (β2-MG), immunoglobulin G (IgG), tumor 15 necrosis factor-α (TNF-α), and glutathione reductase (GR) activities. On the other hand, ANSA176 supplementated diet inhibited or attenuated the OTA-induced damages.
The specific study could be appropriate to be accepted for publication after the application of the following modifications:
1. In the introduction section please add 1-2 paragraphs containing more information regarding the bioelimination biological strategies and bacteria used. In addition, more information about the selection of the bacteria should be added.
2. Page 5, Line 76: please correct the typo at the word specie
3. Please rephrase some parts of the discussion section for plagiarism.
4. Page 10, Line 173: Please add a paragraph with more information in previous studies regarding the putative palatability effects of the bacteria supplements in feed diets.
5. Section 5.1.1.: Please rephrase the whole paragraph
6. Section 5.1.3., Line 218: The centrifugation was performed for 12000 r for 4 min. What is r?
7. Section 5.1.4.: Please rephrase lines 228-230
8. Section 5.2.2.: At the initial experimental design why the authors did not add another treatment with plain A. faecalis ANSA176?
9. Section 5.2.3.: Please rephrase the whole paragraph.
10. Section 5.2.4.: Please describe more analytically the extraction and clean-up procedure.
11. Reference 36 is completely outdated. Please add another more recent publication.
Reviewer 3 Report
The manuscript "Alcaligenes faecalis ANSA176 could alleviate the immune injury and inflammation induced by ochratoxin A" is an important study for biotechnology.
The article is well written and illustrated, however I would suggest splitting figure 1 by isolating the Phylogenetic tree into a separate figure.
The results are clear and detailed. I have only a small remark: The authors begin the description of the results with the phrase "Among the screened bacteria....". At the same time, it is not written anywhere about the screening of different strains.
In my opinion, the article can be published after minor editing.
Round 2
Reviewer 1 Report
The manuscript can be accept in present form.